# Synthesis and Evaluation of Antidepressant Activities of 5-Aryl-4,5-dihydrotetrazolo [1,5-*a*]thieno[2,3-*e*]pyridine Derivatives

**DOI:** 10.3390/molecules24101857

**Published:** 2019-05-14

**Authors:** Shiben Wang, Hui Liu, Xuekun Wang, Kang Lei, Guangyong Li, Zheshan Quan

**Affiliations:** 1College of Pharmacy, Liaocheng University, Liaocheng 252059, China; xuekunwang0610@126.com (X.W.); leikang@lcu.edu.cn (K.L.); liguangyong@lcu.edu.cn (G.L.); 2College of Life Sciences, Liaocheng University, Liaocheng 252059, China; yshxiaohui@163.com; 3College of Pharmacy, Yanbian University, Yanji 133002, China; quickly6688@163.com

**Keywords:** synthesis, antidepressant, FST, 5-HT, molecular docking studies

## Abstract

In this study, we synthetized a series of 5-aryl-4,5-dihydrotetrazolo[1,5-*a*]thieno[2,3-*e*]pyridine derivatives containing tetrazole and other heterocycle substituents, i.e., triazole, methyltriazole, and triazolone. The forced swim test (FST) and tail suspension test (TST) were used to evaluate the antidepressant activity of the target compounds. The compound 5-[4-(trifluoromethyl)phenyl]-4,5-dihydrotetrazolo[1,5-*a*]thieno[2,3-*e*]pyridine (**4i**) showed the highest antidepressant activity, with a reduced immobility time of 55.33% when compared with the control group. Using an open-field test, compound **4i** was shown to not affect spontaneous activity of mice. The determination of in vivo 5-hydroxytryptamine (5-HT) concentration showed that compound **4i** may have an effect in the mouse brain. The biological activities of all synthetized compounds were verified by molecular docking studies. Compound **4i** showed significant interactions with residues of the 5-HT_1A_ receptor homology model.

## 1. Introduction

The structure of the thiophene moiety has been gaining much attention owing to its significant influence on biological activity, showing anti-inflammatory [1], antibacterial [2], anti-anxiety [3], antitumor [4,5], and antidepressant effects [6]. It is therefore essential to further investigate derivatives with a thiophene structure.

It is well-known that in compounds with a tetrazole fragment, this fragment has a great influence on the biological activity of the compound, showing, for instance, antifungal [7], antibacterial [8], antileishmanial [9], anticancer [10,11], anticonvulsant [12,13], and antidepressant effects [14]. Our group has studied a series of 7-alkyl-7*H*-tetrazolo [1,5-*g*]purine derivatives (**I**) (Figure 1), where most of the target compounds showed a significant antidepressant activity [15]. Paroxetine is primarily used for the treatment of depression; the 4-arylpiperidine fragment of this drug has a great influence on its antidepressant activity. In this study, we designed and synthesized a series of 5-aryl-4,5-dihydrotetrazolo[1,5-*a*]thieno[2,3-*e*]pyridine (**4a**–**p**) derivatives by combining biologically active structural fragments such as tetrazole, thiophene ring, and 4-phenylpiperidine. To investigate the effect of the tetrazole fragment on antidepressant activity, the tetrazole ring was replaced, following the bioisosteric principle, by other heterocycles such as triazole, methyltriazole, and triazolone to obtain the target compounds **5**, **6**, and **7**, respectively.

The forced swim test (FST) was used to evaluate the antidepressant activity of the target compounds. Moreover, the most active compound was further tested using a tail suspension test (TST). The concentration of 5-hydroxytyramine (5-HT) was determined in mouse brain to explore the possible mechanism of action of compound **4i**. A homology model of the 5-HT_1A_ receptor was built using Discovery Studio software and used to perform a molecular docking study. The synthesized target compounds were characterized using infrared spectroscopy, high resolution mass spectrometry, and ^1^H-nuclear/^13^C-nuclear magnetic resonance spectroscopy techniques.

## 2. Results and Discussion

### 2.1. Chemistry

The synthesis of the target compounds is outlined in Scheme 1. Compound **1a**–**p** [16] (differently substituted on the benzene ring using electron withdrawing such as Cl, F, CF_3_, Br, 2-Cl-6-F or electron donating such as OCH_3_, 3,4,5-OCH_3_, as well as unsubstituted on the benzene ring) was mixed with P_2_S_5_ in acetonitrile in the presence of triethylamine to obtain the intermediate **2a**–**p**. Compound **3a**–**p** can be easily obtained with a high yield under reflux conditions after the reaction of compound **2a**–**p** and hydrazine hydrate with ethanol as a solvent, where ethanol can also be replaced by methanol or tetrahydrofuran [17]. The target compounds, 7-phenyl-4,5,6,7-tetrahydrothieno[3,2-*b*]pyridine derivatives containing tetrazole, triazole, methyltriazole, and triazolone (compounds **4a**–**p**, **5**, **6**, and **7**, respectively), were synthesized by mixing compound **3a**–**p** with NaNO_2_ (i.e., 5% HCl solution cooled in ice water, withNaNO_2_ slowly added to prevent the increase of by-products caused by excessive drip acceleration), HC(OC_2_H_5_)_3_ (that acts as both a solvent and a reactant at 100 °C), CH_3_COOH (that acts as both a solvent and a reactant, and a high yield can be obtained by refluxing), and NH_2_CONH_2_ (the temperature must be controlled at around 170 °C), respectively [18]. The structures of the target compounds were characterized using spectral methods (see Appendix A).

### 2.2. Pharmacology

#### 2.2.1. Forced Swim Test (FST)

The FST is the most direct and effective method for screening antidepressants. A variety of antidepressant drugs have been developed using this method to evaluate their efficacy. In this study, we used FST as a preliminary antidepressant activity test for all target compounds. Fluoxetine, a widely used selective 5-HT reuptake inhibitor (SSRI), selectively inhibits 5-HT transporters and blocks pre-synaptic membrane reuptake of 5-HT. This extends and increases the effects of 5-HT, resulting in antidepressant effects. In these experiments, we used fluoxetine as a positive control, as already reported in many studies [19]. Results obtained for FST and antidepressant activity are shown in Table 1.

As illustrated in Table 1, most of the compounds in the **4a**–**p** series showed a better antidepressant activity (40 mg/kg, intraperitoneal injection) when compared with the control group. Among those, compounds **4a**, **4d**–**e**, **4k**, and **4m**–**n** showed a significant difference when compared with the control (0.01 < *p* < 0.05). Furthermore, the six compounds **4c**, **4f**–**i**, and **4l** showed a substantial significant difference when compared with the control group (*p* < 0.01). Overall, compound **4i** demonstrated the best antidepressant activity and compound **4i** led to a mice immobility time of 51.43 s, which was similar to that of fluoxetine, i.e., 54.21 s. In a second step, the most active compound **4i** was structurally modified by replacing the biologically active tetrazole fragment with other azole moieties (i.e., triazole, methyltriazole, and triazolone), to potentially synthetize a more active compound. The antidepressant activity of compounds **5**, **6** and **7** was found to be lower than that of compound **4i**. The structure–activity relationship was obtained based on the pharmacological results shown in Table 1. Compounds containing electron-withdrawing groups (i.e., Cl, F, and CF_3_) showed an antidepressant activity order of *p*-Cl > *m*-Cl > *o*-Cl, *p*-F > *m*-F > *o*-F, *p*-CF_3_ > *o*-CF_3_ > *m*-CF_3_. This indicates that a compound having an electron-withdrawing group substituted at its *p*-position shows a better activity than a substitution in the *o* and *m*-positions. On the contrary, in the presence of the electron-donating group OCH_3_, the activity order was *p*-OCH_3_ > *m*-OCH_3_ > *o*-OCH_3_, while substitution in the *p*-position also led to the highest compound activity. The most active compound **4i** was found to be more active than compounds **5**, **6**, and **7**, with an activity order of **4i** > **7** > **5** > **6**.

Compound **4i** and fluoxetine were further tested to evaluate their antidepressant activity at different doses, i.e., 10, 20, and 40 mg/kg. Table 2 shows that compound **4i** and fluoxetine have different antidepressant activities at different doses, with the best antidepressant activity observed with a dose of 40 mg/kg. Therefore, the antidepressant activity of compound **4i** appeared to be dose-dependent, with a gradually increased antidepressant activity observed when increasing the dose.

#### 2.2.2. Tail Suspension Test (TST)

The most active compound **4i** was selected for TST (Table 3) to further confirm the antidepressant activity measured in FST. Fluoxetine was also used as positive control. TST experiments were carried out with a dose of 40 mg/kg for compound **4i** and fluoxetine. As illustrated in Table 3, compound **4i** also demonstrated a significant antidepressant activity during TST. The immobility time of mice was 77.18 s, a substantially significant difference relative to the control group (*p* < 0.01).

#### 2.2.3. Open-Field Test

The open-field test was used to determine whether compound **4i** affects the spontaneous locomotor activity of mice (Figure 2). This test is a classical animal experimental model to assess the autonomic effects of drugs and the general animal activities. Since the reduced immobility time in behavioral despair and depression animal models may be caused by excitation of sympathetic nerves by the drug, the open-field experiment was used to evaluate the central excitability of **4i** [20,21]. Compared with the control group, no significant difference was observed for the compound **4i** (*p* > 0.05, motor activity: crossing, rearing, and grooming). These findings thus exclude any false positive results attributed to central activity excitability.

#### 2.2.4. Determination of 5-HT Concentration

Nowadays, monoamine neurotransmitters such as 5-HT are recognized to play a role in the neurobiochemical mechanisms of depression. The brain is scattered with monoamine neurotransmitter pathways that primarily control physiological activities. Changes in neurotransmitters levels affect the monoamine-based transmitter pathways, resulting in a variety of clinical depressive symptoms. The results of a pathological autopsy of depression showed a decrease in 5-HT levels in the brainstem and frontal lobe, as well as a decrease in the total amount of 5-HT receptors in the hippocampus [22]. In the present study, the effect of compound **4i** on the concentration of 5-HT in mouse brain tissues was determined using enzyme-linked immunosorbent assay (ELISA). The results showed that the concentration of 5-HT in brain tissue in the group treated with compound **4i** and the fluoxetine group (40 mg/kg) was significantly higher than that of the control group (Table 4).

#### 2.2.5. Docking Study

Molecular docking is an important means to explore the possible mechanisms of biologically active compounds. The 5-HT_1A_ receptor plays a role in the pathogenesis of various mental and neurological diseases. Activation of postsynaptic 5-HT_1A_ receptors is important for an adequate response to antidepressants [23,24]. Here, we used molecular docking to investigate the interaction between compound **4i** and different residues of the 5-HT_1A_ receptor homology model. The docking results are shown in Figure 3.

According to existing literature [25], the main amino acid residues present at the active site of the 5-HT_1A_ receptor homology model are Ala93, Ala263, Ala365, Ala383, Asn386, Asp116, Cys120, Thr379, Gln97, Gly382, Ile113, Ile124, Ile167, Phe112, Phe361, Phe362, Ser199, Thr196, Thr121, Thr200, and Val117. Molecular docking showed that compound **4i** interacts with amino acid residues present at the 5-HT_1A_ receptor active site, for instance via hydrogen bond interaction between the N atom in the triazole fragment and the Thr196 residue. Furthermore, at the site of the active pocket of the 5-HT_1A_ receptor, compound **4i** can form a hydrophobic and hydrogen bond interaction with the amino acid residues surrounding this active pocket. Therefore, the mode of action of compound **4i** in exerting antidepressant activity may be closely related to its interaction with the 5-HT_1A_ receptor.

## 3. Experimental Section

### 3.1. Chemistry

An X-4 binocular microscope melting point apparatus (Ningbo Biocotek scientific equipment Co., Ltd, Ningbo, China) was used to determine the melting points (Mp). All NMR (^1^H and ^13^C NMR) experiments were performed using an AV-300 spectrometer (Bruker Bioscience, Billerica, MA, USA), with CDCl_3_ or DMSO-*d*_6_ as solvent, as well as tetramethylsilane as internal standard. Infrared (IR) spectra of the target compounds were obtained with an IR Prestige-21 instrument (in KBr, Shimadzu, Tokyo, Japan). High resolution mass spectra (HRMS) were obtained with mass spectrometry (Bruker Daltonik GmbH, Bremen, Germany). Most chemicals used for the synthesis were commercial products without further purification. The synthesis of starting material **1a**–**p** was based on the synthesis method of Zhang et al. [16].

#### 3.1.1. Synthesis of 7-substitutedphenyl-6,7-dihydrothieno[3,2-*b*]pyridine-5(4*H*)-thiones (**2a**–**p**)

A hundred milliliters of a solvent mixture composed of acetonitrile/triethylamine (3:1) was added to a 250 mL round bottom flask, and P_2_S_5_ (7.5 mmol) was added in small portions in an ice bath. The intermediate compound **1a**–**p** (5.0 mmol) was then added to the reaction solution followed by reflux for 4–7 h. After completion of the reaction, the solvent was evaporated under reduced pressure. Dichloromethane (50 mL) was then added to the solid residue and washed three times with water. The dichloromethane layer was subsequently dried over anhydrous magnesium sulfate, filtered and evaporated, producing the white crude compound **2a**–**p**.

#### 3.1.2. Synthesis of 5-hydrazono-7-substitutedphenyl-4,5,6,7-tetrahydrothieno[3,2-*b*]pyridines (**3a**–**p**)

Compound **2a**–**p** (3.0 mmol) and hydrazine hydrate (6.0 mmol) were added to a round bottom flask containing 20 mL of methanol. The reaction took place at 70 °C for 1 h. After the reaction was completed, most of the methanol was removed under reduced pressure and cooled. Next, a small amount of cold methanol was used for filtration and rinsing, leading to a pale yellow or white solid.

#### 3.1.3. Synthesis of 5-aryl-4,5-dihydrotetrazolo[1,5-*a*]thieno[2,3-*e*]pyridine (**4a**–**p**)

Selected intermediates from compound **3a**–**p** (3.0 mmol) were added to a round bottom flask containing 10 mL of 5% HCl, and the mixture was cooled in ice water. An aqueous solution of NaNO_2_ was then slowly added. The reaction was monitored by thin liquid chromatography (TLC). After completion of the reaction, the crude product was obtained by suction filtration under reduced pressure. Purification was performed using column chromatography (methanol/dichloromethane = 1/80) to give a white solid.

5-(2-Chlorophenyl)-4,5-dihydrotetrazolo[1,5-*a*]thieno[2,3-*e*]pyridine (**4a**): Yield 68%. Mp: 159–160 °C. ^1^H-NMR (CDCl_3_, 300 MHz, ppm) *δ*: 3.59–3.82 (m, 2H, CH_2_), 5.18 (t, 1H, *J* = 7.4 Hz, CH), 6.89–7.28 (m, 4H, Ar-H), 7.45 (d, 1H, *J* = 5.3 Hz, S-C=C-H), 7.81 (d, 1H, *J* = 5.3 Hz, S-C-H). ^13^C-NMR (CDCl_3_, 75 MHz) *δ*: 27.5, 35.9, 117.5, 127.2, 127.7, 128.2, 128.6, 129.5, 130.3, 131.3, 133.0, 137.6, 149.5. IR (KBr, cm^−1^): 1479 (C=N). ESI-HRMS calcd for C_13_H_10_ClN_4_S^+^ ([M + H]^+^): 289.0309; found: 289.0302.

5-(3-Chlorophenyl)-4,5-dihydrotetrazolo[1,5-*a*]thieno[2,3-*e*]pyridine (**4b**): Yield 77%. Mp: 169–170 °C. ^1^H-NMR (CDCl_3_, 300 MHz, ppm) *δ*: 3.47–3.82 (m, 2H, CH_2_), 4.61 (t, 1H, *J* = 8.4 Hz, CH), 7.10–7.32 (m, 4H, Ar-H), 7.43 (d, 1H, *J* = 5.3 Hz, S-C=C-H), 7.58 (d, 1H, *J* = 5.3 Hz, S-C-H). ^13^C-NMR (CDCl_3_, 75 MHz) *δ*: 29.0, 39.4, 117.6, 125.6, 127.2, 127.6, 128.7, 129.5, 130.6, 130.7, 135.1, 142.3, 149.6. IR (KBr, cm^−1^): 1479 (C=N). ESI-HRMS calcd for C_13_H_10_ClN_4_S^+^ ([M + H]^+^): 289.0309; found:289.0305.

5-(4-Chlorophenyl)-4,5-dihydrotetrazolo[1,5-*a*]thieno[2,3-*e*]pyridine (**4c**): Yield 72%. Mp: 108–110 °C. ^1^H-NMR (CDCl_3_, 300 MHz, ppm) *δ*: 3.44–3.80 (m, 2H, CH_2_), 4.61 (t, 1H, *J* = 8.5 Hz, CH), 7.16 (d, 2H, *J* = 4.1 Hz, Ar-H), 7.35 (d, 2H, *J* = 4.2 Hz, Ar-H), 7.41 (d, 1H, *J* = 5.4 Hz, S-C=C-H), 7.56 (d, 1H, *J* = 5.4 Hz, S-C-H). ^13^C-NMR (CDCl_3_, 75 MHz) *δ*: 29.1, 39.2, 117.6, 127.1, 128.8, 128.8, 129.5, 129.5, 129.9, 130.7, 134.3, 138.8, 149.7. IR (KBr, cm^−1^): 1479 (C=N). ESI-HRMS calcd for C_13_H_10_ClN_4_S^+^ ([M + H]^+^): 289.0309; found: 289.0303.

5-(2-Flulorophenyl)-4,5-dihydrotetrazolo[1,5-*a*]thieno[2,3-*e*]pyridine (**4d**): Yield 62%. Mp: 122–123 °C. ^1^H-NMR (CDCl_3_, 300 MHz, ppm) *δ*: 3.60–3.81 (m, 2H, CH_2_), 4.97 (t, 1H, *J* = 7.7 Hz, CH), 6.92–7.36 (m, 4H, Ar-H), 7.43 (d, 1H, *J* = 5.4 Hz, S-C=C-H), 7.59 (d, 1H, *J* = 5.4 Hz, S-C-H). ^13^C-NMR (CDCl_3_, 75 MHz) *δ*: 27.6, 32.8, 116.3, 117.6, 124.9, 126.9, 127.3, 128.4, 128.7, 130.2, 130.9, 149.7, 158.4, 161.7. IR (KBr, cm^−1^): 1485 (C=N). ESI-HRMS calcd for C_13_H_10_FN_4_S^+^ ([M + H]^+^): 273.0605; found: 273.0609.

5-(3-Flulorophenyl)-4,5-dihydrotetrazolo[1,5-*a*]thieno[2,3-*e*]pyridine (**4e**): Yield 61%. Mp: 124–125 °C. ^1^H-NMR (CDCl_3_, 300 MHz, ppm) *δ*: 3.49–3.83 (m, 2H, CH_2_), 4.63 (t, 1H, *J* = 8.3 Hz, CH), 6.91–7.39 (m, 4H, Ar-H), 7.43 (d, 1H, *J* = 5.4 Hz, S-C=C-H), 7.59 (d, 1H, *J* = 5.4 Hz, S-C-H). ^13^C-NMR (CDCl_3_, 75 MHz) *δ*: 29.0, 39.4, 114.6, 115.6, 117.6, 123.1, 127.1, 129.6, 130.7, 131.0, 142.7, 149.7, 161.4, 164.7. IR (KBr, cm^−1^): 1485 (C=N). ESI-HRMS calcd for C_13_H_10_FN_4_S^+^ ([M + H]^+^): 273.0605; found: 273.0611.

5-(4-Flulorophenyl)-4,5-dihydrotetrazolo[1,5-*a*]thieno[2,3-*e*]pyridine (**4f**): Yield 55%. Mp: 123–124 °C. ^1^H-NMR (CDCl_3_, 300 MHz, ppm) *δ*: 3.44–3.80 (m, 2H, CH_2_), 4.62 (t, 1H, *J* = 8.5 Hz, CH), 7.03–7.27 (m, 4H, Ar-H), 7.41 (d, 1H, *J* = 5.3 Hz, S-C=C-H), 7.56 (d, 1H, *J* = 5.3 Hz, S-C-H). ^13^C-NMR (CDCl_3_, 75 MHz) *δ*: 29.3, 39.1, 116.1, 116.4, 117.6, 126.9, 129.0, 129.1, 130.4, 130.6, 136.0, 149.8, 160.8, 164.1. IR (KBr, cm^−1^): 1485 (C=N). ESI-HRMS calcd for C_13_H_10_FN_4_S^+^ ([M + H]^+^): 273.0605; found: 273.0601.

5-[2-(Trifluoromethyl)phenyl]-4,5-dihydrotetrazolo[1,5-*a*]thieno[2,3-*e*]pyridine (**4g**): Yield 51%. Mp: 201–202 °C. ^1^H-NMR (CDCl_3_, 300 MHz, ppm) *δ*: 3.44–3.90 (m, 2H, CH_2_), 5.08 (t, 1H, *J* = 8.4 Hz, CH), 7.24–7.78 (m, 4H, Ar-H), 7.44 (d, 1H, *J* = 5.4 Hz, S-C=C-H), 7.61 (d, 1H, *J* = 5.4 Hz, S-C-H). ^13^C-NMR (CDCl_3_, 75 MHz) *δ*: 29.3, 35.3, 117.5, 125.9, 126.4, 127.4, 128.0, 128.4, 129.2, 129.7, 131.2, 132.9, 139.6, 149.4. IR (KBr, cm^−1^): 1485 (C=N). ESI-HRMS calcd for C_14_H_10_F_3_N_4_S^+^ ([M + H]^+^): 323.0573; found: 323.0577.

5-[3-(Trifluoromethyl)phenyl]-4,5-dihydrotetrazolo[1,5-*a*]thieno[2,3-*e*]pyridine (**4h**): Yield 53%. Mp: 141–142 °C. ^1^H-NMR (CDCl_3_, 300 MHz, ppm) *δ*: 3.49–3.87 (m, 2H, CH_2_), 4.72 (t, 1H, *J* = 8.5 Hz, CH), 7.40–7.64 (m, 4H, Ar-H), 7.44 (d, 1H, *J* = 5.4 Hz, S-C=C-H), 7.59 (d, 1H, *J* = 5.4 Hz, S-C-H). ^13^C-NMR (CDCl_3_, 75 MHz) *δ*: 29.1, 39.6, 117.7, 121.9, 124.4, 125.4, 127.3, 129.3, 129.9, 130.8, 130.9, 131.9, 141.3, 149.6. IR (KBr, cm^−1^): 1485 (C=N). ESI-HRMS calcd for C_14_H_10_F_3_N_4_S^+^ ([M + H]^+^): 323.0573; found: 323.0579.

5-[4-(Trifluoromethyl)phenyl]-4,5-dihydrotetrazolo[1,5-*a*]thieno[2,3-*e*]pyridine (**4i**): Yield 48%. Mp: 155–156 °C. ^1^H-NMR (CDCl_3_, 300 MHz, ppm) *δ*: 3.50–3.86 (m, 2H, CH_2_), 4.72 (t, 1H, *J* = 8.3 Hz, CH), 7.36 (d, 2H, *J* = 7.9 Hz, Ar-H), 7.45 (d, 1H, *J* = 5.4 Hz, S-C=C-H), 7.60 (d, 1H, *J* = 5.4 Hz, S-C-H), 7.66 (d, 2H, *J* = 8.0 Hz, Ar-H). ^13^C-NMR (CDCl_3_, 75 MHz) *δ*: 29.0, 39.5, 117.7, 126.4, 127.3, 127.3, 127.8, 127.8, 127.8, 127.8, 129.2, 130.9, 144.3, 149.5. IR (KBr, cm^−1^): 1485 (C=N). ESI-HRMS calcd for C_14_H_10_F_3_N_4_S^+^ ([M + H]^+^): 323.0573; found: 323.0577.

5-(2-Methoxyphenyl)-4,5-dihydrotetrazolo[1,5-*a*]thieno[2,3-*e*]pyridine (**4j**): Yield 42%. Mp: 139–141 °C. ^1^H-NMR (CDCl_3_, 300 MHz, ppm) *δ*: 3.67 (d, 2H, *J* = 7.2 Hz, CH_2_), 3.82 (s, 3H, OCH_3_), 5.02 (t, 1H, *J* = 7.2 Hz, CH), 6.85–7.30 (m, 4H, Ar-H), 7.38 (d, 1H, *J* = 5.4 Hz, S-C=C-H), 7.56 (d, 1H, *J* = 5.4 Hz, S-C-H). ^13^C-NMR (CDCl_3_, 75 MHz) *δ*: 26.9, 33.5, 55.4, 110.9, 117.3, 120.9, 126.3, 127.5, 128.5, 129.3, 129.9, 130.7, 150.3, 156.4. IR (KBr, cm^−1^): 1510 (C=N). ESI-HRMS calcd for C_14_H_13_N_4_OS^+^ ([M + H]^+^): 285.0805; found: 285.0810.

5-(3-Methoxyphenyl)-4,5-dihydrotetrazolo[1,5-*a*]thieno[2,3-*e*]pyridine (**4k**): Yield 66%. Mp: 104–105 °C. ^1^H-NMR (CDCl_3_, 300 MHz, ppm) *δ*: 3.49–3.82 (m, 2H, CH_2_), 3.80 (s, 3H, OCH_3_), 4.58 (t, 1H, *J* = 8.5 Hz, CH), 6.78–7.33 (m, 4H, Ar-H), 7.40 (d, 1H, *J* = 5.4 Hz, S-C=C-H), 7.58 (d, 1H, *J* = 5.4 Hz, S-C-H). ^13^C-NMR (CDCl_3_, 75 MHz) *δ*: 29.1, 39.8, 55.3, 113.3, 113.5, 115.1, 117.5, 119.5, 126.8, 130.4, 130.5, 141.8, 149.9, 160.2. IR (KBr, cm^−1^): 1510 (C=N). ESI-HRMS calcd for C_14_H_13_N_4_OS^+^ ([M + H]^+^): 285.0805; found: 285.0809.

5-(4-Methoxyphenyl)-4,5-dihydrotetrazolo[1,5-*a*]thieno[2,3-*e*]pyridine (**4l**): Yield 60%. Mp: 156–157 °C. ^1^H-NMR (CDCl_3_, 300 MHz, ppm) *δ*: 3.43–3.79 (m, 2H, CH_2_), 3.81 (s, 3H, OCH_3_), 4.56 (t, 1H, *J* = 8.7 Hz, CH), 6.89 (d, 2H, *J* = 8.4 Hz, Ar-H), 7.18 (d, 2H, *J* = 8.3 Hz, Ar-H), 7.38 (d, 1H, *J* = 5.3 Hz, S-C=C-H), 7.56 (d, 1H, *J* = 5.2 Hz, S-C-H). ^13^C-NMR (CDCl_3_, 75 MHz) *δ*: 29.3, 39.1, 55.3, 114.6, 114.6, 117.6, 126.7, 128.5, 128.5, 130.4, 131.4, 132.2, 150.1, 159.5. IR (KBr, cm^−1^): 1510 (C=N). ESI-HRMS calcd for C_14_H_13_N_4_OS^+^ ([M + H]^+^): 285.0805; found: 285.0811.

5-(4-Bromophenyl)-4,5-dihydrotetrazolo[1,5-*a*]thieno[2,3-*e*]pyridine (**4m**): Yield 56%. Mp: 119–121 °C. ^1^H-NMR (CDCl_3_, 300 MHz, ppm) *δ*: 3.43-3.80 (m, 2H, CH_2_), 4.59 (t, 1H, *J* = 8.4 Hz, CH), 7.10 (d, 2H, *J* = 8.3 Hz, Ar-H), 7.41 (d, 1H, *J* = 5.4 Hz, S-C=C-H), 7.50 (d, 2H, *J* = 8.2 Hz, Ar-H), 7.56 (d, 1H, *J* = 5.40 Hz, S-C-H). ^13^C-NMR (CDCl_3_, 75 MHz) *δ*: 29.0, 39.3, 50.8, 117.6, 122.4, 127.1, 129.1, 129.1, 129.9, 130.7, 132.5, 132.5, 139.3, 149.7. IR (KBr, cm^−1^): 1500 (C=N). ESI-HRMS calcd for C_13_H_10_BrN_4_S^+^ ([M + H]^+^): 322.9804; found: 322.9809.

5-Phenyl-4,5-dihydrotetrazolo[1,5-*a*]thieno[2,3-*e*]pyridine (**4n**): Yield 68%. Mp: 101–102 °C. ^1^H-NMR (CDCl_3_, 300 MHz, ppm) *δ*: 3.48–3.81 (m, 2H, CH_2_), 4.61 (t, 1H, *J* = 8.5 Hz, CH), 7.23–7.38 (m, 5H, Ar-H), 7.39 (d, 1H, *J* = 5.4 Hz, S-C=C-H), 7.57 (d, 1H, *J* = 5.3 Hz, S-C-H). ^13^C-NMR (CDCl_3_, 75 MHz) *δ*: 29.1, 39.8, 117.5, 126.8, 127.4, 127.4, 128.4, 129.3, 129.3, 130.5, 130.7, 140.3, 149.9. IR (KBr, cm^−1^): 1481 (C=N). ESI-HRMS calcd for C_13_H_11_N_4_S^+^ ([M + H]^+^): 255.0699; found: 255.0693.

5-(2-Chloro-6-fluorophenyl)-4,5-dihydrotetrazolo[1,5-*a*]thieno[2,3-*e*]pyridine (**4o**): Yield 54%. Mp: 179–180 °C. ^1^H-NMR (CDCl_3_, 300 MHz, ppm) *δ*: 3.76–3.84 (m, 2H, CH_2_), 5.36 (t, 1H, *J* = 9.3 Hz, CH), 7.04–7.32 (m, 3H, Ar-H), 7.35 (d, 1H, *J* = 5.5 Hz, S-C=C-H), 7.54 (d, 1H, *J* = 5.3 Hz, S-C-H). ^13^C-NMR (CDCl_3_, 75 MHz) *δ*: 25.4, 50.8, 115.4, 117.5, 117.5, 126.2, 126.2, 128.5, 129.9, 130.4, 130.5, 134.6, 149.9. IR (KBr, cm^−1^): 1485 (C=N). ESI-HRMS calcd for C_13_H_9_ClFN_4_S^+^ ([M + H]^+^): 307.0215; found: 307.0209.

5-(3,4,5-Trimethoxyphenyl)-4,5-dihydrotetrazolo[1,5-*a*]thieno[2,3-*e*]pyridine (**4p**): Yield 62%. Mp: 200–201 °C. ^1^H-NMR (CDCl_3_, 300 MHz, ppm) *δ*: 3.47–3.85 (m, 2H, CH_2_), 3.77–3.85 (m, 9H, (OCH_3_)_3_), 4.53 (t, 1H, *J* = 8.6 Hz, CH), 6.45 (s, 2H, Ar-H), 7.41 (d, 1H, *J* = 5.4 Hz, S-C=C-H), 7.57 (d, 1H, *J* = 5.4 Hz, S-C-H). ^13^C-NMR (CDCl_3_, 75 MHz) *δ*: 29.3, 40.2, 50.8, 56.2, 60.9, 104.4, 104.4, 117.5, 126.9, 130.6, 130.7, 135.7, 137.9, 150.0, 153.8, 153.8. IR (KBr, cm^−1^): 1510 (C=N). ESI-HRMS calcd for C_16_H_17_N_4_O_3_S^+^ ([M + H]^+^): 345.1016; found: 345.1021.

#### 3.1.4. Synthesis of 5-[4-(trifluoromethyl)phenyl]-4,5-dihydrothieno[2,3-*e*][1,2,4]triazolo[4,3-*a*]pyridine (**5**)

The intermediate **3i** (3.0 mmol) and 10 mL of triethyl orthoformate were mixed into a 50 mL round bottom flask, and then reacted at 100 °C for 6 h. After completion of the reaction, the triethyl orthoformate was removed under reduced pressure using a vacuum pump. Thirty milliliters of water were added to obtain a solid residue, which was then filtered and dried to give a crude product. Purification was performed using column chromatography (methanol/dichloromethane = 1/100) to give a white solid.

5-[4-(Trifluoromethyl)phenyl]-4,5-dihydrothieno[2,3-*e*][1,2,4]triazolo[4,3-*a*]pyridine (**5**): Yield 61%. Mp: 158–160 °C. ^1^H-NMR (CDCl_3_, 300 MHz, ppm) *δ*: 3.40–3.71 (m, 2H, CH_2_), 4.59 (t, 1H, *J* = 7.8 Hz, CH), 7.22–7.23 (d, 1H, S-C=C-H), 7.35–7.64 (m, 4H, Ar-H), 7.38–7.40 (d, 1H, S-C-H), 8.52 (s, 1H, Triazole-H). ^13^C-NMR (CDCl_3_, 75 MHz) *δ*: 29.7, 39.6, 116.8, 126.1, 126.2, 126.7, 127.9, 128.1, 128.1, 130.2, 130.2, 130.6, 137.6, 144.9, 148.9. IR (KBr, cm^−1^): 1502 (C=N). ESI-HRMS calcd for C_15_H_11_F_3_N_3_S^+^ ([M + H]^+^): 322.0620; found: 322.0626.

#### 3.1.5. Synthesis of 1-methyl-5-[4-(trifluoromethyl)phenyl]-4,5-dihydrothieno[2,3-*e*][1,2,4]triazolo[4,3-*a*]pyridine (**6**)

The selected intermediates from compound **3i** synthesis (3.0 mmol) were added to a round bottom flask containing 15 mL of acetic acid and reacted at 140 °C for 2 h. After the reaction was completed, acetic acid was removed under reduced pressure. Thirty milliliters of water were added, filtered, and dried to give a crude product. Purification was performed using column chromatography (methanol/dichloromethane = 1/100) to give a white solid.

1-Methyl-5-[4-(trifluoromethyl)phenyl]-4,5-dihydrothieno[2,3-*e*][1,2,4]triazolo[4,3-*a*]pyridine (**6**): Yield 48%. Mp: 117–119 °C. ^1^H-NMR (CDCl_3_, 300 MHz, ppm) *δ*: 2.76 (s, 3H, CH_3_), 3.41–3.60 (m, 2H, CH_2_), 4.54 (s, 1H, CH), 7.32–7.63 (m, 4H, Ar-H), 7.38 (s, 1H, S-C=C-H), 7.62 (d, 1H, *J* = 5.5 Hz, S-C-H). ^13^C-NMR (CDCl_3_, 75 MHz) *δ*: 12.5, 29.9, 39.7, 117.0, 122.1, 126.0, 126.1, 126.1, 127.9, 127.9, 127.9, 129.4, 130.2, 131.0, 144.7, 175.2. IR (KBr, cm^−1^): 1666 (C=N). ESI-HRMS calcd for C_16_H_13_F_3_N_3_S^+^ ([M + H]^+^): 336.0777; found: 336.0785.

#### 3.1.6. Synthesis of 5-[4-(trifluoromethyl)phenyl]-4,5-dihydrothieno[2,3-*e*][1,2,4]triazolo[4,3-*a*]pyridin-1(2*H*)-one (**7**)

Selected intermediates from **3i** (5.0 mmol) and urea (10.0 mmol) were added into a 50 mL round bottom flask, and the mixture was heated to 170 °C for reaction for 4 h. After the reaction was completed, 30 mL of water was added, suction filtered, and dried to give a crude product. Purification was performed using column chromatography (methanol/dichloromethane = 1/60) to give a white solid. The yield, melting point and nuclear magnetic data of the target compounds are shown below.

5-[4-(Trifluoromethyl)phenyl]-4,5-dihydrothieno[2,3-*e*][1,2,4]triazolo[4,3-*a*]pyridin-1(2*H*)-one (**7**): Yield 62%. Mp: 229–231 °C. ^1^H-NMR (CDCl_3_, 300 MHz, ppm) *δ*: 3.19–3.35 (m, 2H, CH_2_), 4.55 (t, 1H, *J* = 7.6 Hz, CH), 7.29–7.64 (m, 4H, Ar-H), 7.29–7.81 (m, 2H, S-C=C-H and S-C-H), 9.38 (s, 1H, CONH). ^13^C-NMR (CDCl_3_, 75 MHz) *δ*: 29.9, 38.1, 117.4, 124.6, 125.5, 126.3, 126.5, 128.7, 128.7, 128.7, 128.7, 130.8, 142.5, 147.2, 151.5. IR (KBr, cm^−1^): 1707 (C=O). ESI-HRMS calcd for C_15_H_11_F_3_N_3_OS^+^ ([M + H]^+^): 338.0569; found: 338.0573.

### 3.2. Pharmacology

All target compounds were evaluated in vivo in Kunming mice (18–22 g). Mice had free access to food and water before the experiments. Target compounds were dissolved in dimethyl sulfoxide or 0.5% methylcellulose. Fluoxetine was used as positive control. Two experimental models, FST and TST, were used to estimate the antidepressant activity of target compounds. The 5-HT enzyme-linked immunosorbent assay (ELISA) kits were used to determine the concentrations of 5-HT in mouse tissues.

#### 3.2.1. Evaluation of Antidepressant Activity

As reported in the literature, antidepressant activities of the target compounds were determined using FST [26,27]. Based on the preliminary FST screening results, the potent active compound **4i** was selected to further evaluate its antidepressant activity using TST [28,29]. Next, the highly active compound **4i** was assessed for its activity on animals using an open-field test [30]. 5-HT concentration was measured using the ELISA method to determine whether compound **4i** showed an effect in the mouse brain.

#### 3.2.2. Molecular Docking Studies

The sequence of the 5-HT_1A_ receptor was retrieved from Research Collaboratory for Structural Bioinformatics (RCSB) in Protein Data Bank (PDB) (web site: https://www.rcsb.org/). Structure crystals with high homology (PDB: 4IAR, 4IAQ, 5V54, and 6G79) were used to construct a homology model of the 5-HT_1A_ receptor using Discovery Studio software. The structures of the ligands were sketched using ChemBioDraw Ultra 14.0. LibDock protocol was followed and the top LibDock Score was used for further analysis. The docking results were analyzed with Discovery Studio.

### 3.3. Statistical Analysis

Data were expressed as mean ± standard deviation. One-Way ANOVA analysis of variance was performed using GraphPad Prism 5.0 statistical software. 0.01 < *p* < 0.05 indicated significant difference. *p* < 0.01 indicates that the difference is very significant.

## 4. Conclusions

In the present study, a series of 5-aryl-4,5-dyhidrotetrazolo[1,5-*a*]thieno[2,3-*e*]pyridine derivatives containing tetrazole, triazole, methyltriazole, and triazolone moieties were synthesized. Two experimental methods, FST and TST, were used to evaluate the antidepressant activity of the target compounds. The pharmacological results showed that most of the target compounds exhibited a significant antidepressant activity in FST experiments. Among these compounds, 5-[4-(trifluoromethyl)phenyl]-4,5-dihydrotetrazolo[1,5-*a*]thieno[2,3-*e*]pyridine (**4i**) was found to be the most potent. In the open-field test experiments, compound **4i** did not affect spontaneous activity. The determination of in vivo 5-HT concentrations revealed that compound **4i** may have an effect in the mouse brain. Molecular docking experiments showed that compound **4i** significantly interacted with residues of the 5-HT_1A_ receptor homology model. Therefore, the mode of action of compound **4i** supporting its antidepressant activity may be closely related to the 5-HT_1A_ receptor.

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
