# Peer review of "Synthesis and Evaluation of Antidepressant Activities of 5-Aryl-4,5-dihydrotetrazolo [1,5-a]thieno[2,3-e]pyridine Derivatives"

_molecules, 2019, doi:10.3390/molecules24101857_

Reviewer 1 Report

This manuscript entitled: Synthesis and Evaluation of Antidepressant 3 Activities of 5-Substitutedphenyl-4,5 -dihydro4tetrazolo[1,5-a]thieno[2,3-e]pyridine Derivatives describes the synthesis and evaluation of bio-activitiy of novel type of condensed heteroaromatic compounds. The manuscript is divided into two parts. In the first part the synthetic approach to studied compounds is described. Next, the bio-activity is discussed. The introduction part contains some references to bioactive thiophenes and tetrazoles derivatives. The bio-activities of the prepared substances are compared with fluoxetine. The introduction and results discussion are well assembled so I have no comments to these parts. However, my main concern is directed toward the experimental part because this part is missing key information. In the scheme 1 the synthesis of target compounds 4a-p is described and the prepared compounds are well characterized. However, there are no references for the synthesis of starting compounds 1a-p. The synthesis of compounds 2a-p, and 3a-p is described by general procedures but no analytical data for these compounds are provided. Therefore, I recommend to provide analytical data of compounds 2a-p and 3a-p. Only in the case that the compounds 3a-p were prepared from crude compounds 2a-p the analytical data for compounds 2a-p can be omitted. Moreover, the scheme 1 needs to be updated by changing compound numbers 1,2,3 for 1a-p, 2a-p, 3a-p.

                Overall, the manuscript is aimed to interesting applocation of heterocyclic chemistry, thus I recommend to accept the manuscript for publication in Molecules after addressing all the above mentioned is issues.

Author Response

Dear sir or madam,

   thank you very much for your valuable comments on my article, which is very helpful for my article improvement. I have made any changes or responses to your suggestion, as follows:

Point 1: This manuscript entitled: Synthesis and Evaluation of Antidepressant 3 Activities of 5-Substitutedphenyl-4,5 -dihydro4tetrazolo[1,5-a]thieno[2,3-e]pyridine Derivatives describes the synthesis and evaluation of bio-activitiy of novel type of condensed heteroaromatic compounds. The manuscript is divided into two parts. In the first part the synthetic approach to studied compounds is described. Next, the bio-activity is discussed. The introduction part contains some references to bioactive thiophenes and tetrazoles derivatives. The bio-activities of the prepared substances are compared with fluoxetine. The introduction and results discussion are well assembled so I have no comments to these parts. However, my main concern is directed toward the experimental part because this part is missing key information. In the scheme 1 the synthesis of target compounds 4a-p is described and the prepared compounds are well characterized. However, there are no references for the synthesis of starting compounds 1a-p. The synthesis of compounds 2a-p, and 3a-p is described by general procedures but no analytical data for these compounds are provided. Therefore, I recommend to provide analytical data of compounds 2a-p and 3a-p. Only in the case that the compounds 3a-p were prepared from crude compounds 2a-p the analytical data for compounds 2a-p can be omitted.

 Response 1: Thank you very much for your suggestion, the preparation of the starting material 1a-p was carried out by a similar synthesis method of the reference (Zhang, et al. Chin. J. Org. Chem., 2008, 28, 881-884.). The 1H and 13C spectra and analytical data of compounds 1a-p have been attached to the supplymentary. For the synthesis of the intermediates 2a-p and 3a-p, we have not purified, but after the simple processing, the obtained crude product is directly put into the next step, while the intermediates 2a-p and 3a-p have poor stability for a long time and are easily deteriorated. Therefore, no analytical data can be provided. However, the synthetic method reference similar to the intermediate 2a-p, 3a-p has been added in the 2.1. Chemistry section of Results and Discussion.

 Point 2: Moreover, the scheme 1 needs to be updated by changing compound numbers 1,2,3 for 1a-p, 2a-p, 3a-p. Overall, the manuscript is aimed to interesting applocation of heterocyclic chemistry, thus I recommend to accept the manuscript for publication in Molecules after addressing all the above mentioned is issues.

Response 2: In the paper, compound numbers 1,2,3 have been replaced by 1a-p, 2a-p, 3a-p.

Reviewer 2 Report

In this paper the authors describe the procedure for preparation of new 5-aryl-4,5-dyhidrotetrazolo[1,5-a]thieno[2,3-e]pyridines. All target compounds were synthesized for the first time and were well characterized using spectroscopic methods. Pharmacological tests and computation studies were performed to elucidate the antidepressant activity of these compounds.

I would suggest a native English speaker to check the manuscript throughout as multiple language errors are still present.

I recommend that present manuscript can be accepted after mayor revision. The following comments are addressed:

In the part 2.1 Chemistry it would be appropriate to extended details and provide additional references for the synthesis of intermediates 2a-p and 3a-p as well as for the target compounds 4a-p, 5, 6 and 7.  

Line 71: It would be appropriate to add discussion and reference about Fluoxetine as a positive control.

Line 80: The authors concluded from the Table 1 that compound 4i showed the best activity from the tested compounds. However, compound 4f showed similar activity. It would be convenient to do additional pharmacological tests with this compound (TST, Open-field test and 5-HT concentration determination) as well as docking study.

In the section 3.1 Chemistry the authors should be provide explanation about starting materials (commercial availability or references for their synthesis). After inspection using SciFinder I found that starting materials 1a-p as well as intermediates 2a-p and 3a-p are not known compounds in literature. For these compounds the authors should provide spectral characterizations such as for the target compounds 4a-p, 5, 6 and 7. These data should be add in the manuscript or in the Supporting material.

The following comments should be also addressed:

It is more convenient to use the name 5-aryl-4,5-dyhidrotetrazolo[1,5-a]thieno[2,3-e]pyridine instead 5-substituted phenyl-4,5-dyhidrotetrazolo[1,5-a]thieno[2,3-e]pyridine. These corrections should be applied in lines: 3, 12, 37, 179 and 346.   

Line 15 ─ Delete: The results of...

Line 19 Replace 5-HT with 5-hydroxytriptamine (5-HT).

Line 31 Replace that tetrazole with that compounds with tetrazole.

Line 44 Instead 4-phenylpiperidine write 4-arylpiperidine.

Line 56, 57 Explanation in brackets is not clear!!!

Line 67 It would be more convinient to write subtitle: 2.2.1. Forced swiming test (FST)

Line 84 Delete: To our disappointed...

Line 101 It would be more convinient to write subtitle: 2.2.2. Tail suspension test (TST)

Line 164Instead IRPrestige write IR Prestige (Typographical errors).

Line 168It something wrong with this procedure. It is impossible to add 100 mL of a solvent into a 100 mL round bottom flask. Some correction is necessary.

Line 172Please add reaction time.

Line 172Instead extracted write washed.

Line 178Instead methanol cooled write cold methanol.

Line 181It more appropriate instead and reduce the reaction system to 0-5oC write and cool the mixture in an ice water.

General instruction for interpretation of NMR spectra:

·       Give 1H NMR chemical shifts to two digits after the decimal point. Include number of protons represented by the signal, peak multiplicity, and coupling constants needed. (J italicized, reported with up to one digit after the decimal).

·       Give 13C NMR chemical shifts to one digit after the decimal point, unless an additional digit will help distinguish overlapping peaks.

Line 199, 235, 253, 259wrong interpretation of 1H NMR spectra. Instead a duble of dublet 1,4 disubstituted aromatic compounds have two separated doublets. For example, instead 7.16-7.35 (dd, 4H) it is 7.19 (d, 2H) and 7.35 (d, 2H) with appropriate coupling constants.

Line 240wrong interpretation of 1H NMR spectra. Instead 3.66-3.68 (d, 2H, J = 7.23 Hz, CH2) write 3.67(d, 2H, J = 7.23 Hz, CH2). Also, instead 3.82 (s, 1H, OCH3) write 3.82 (s, 3H, OCH3).

Line 246instead 3.80 (s, 1H, OCH3) write 3.80 (s, 3H, OCH3).

Line 252instead 3.81 (s, 1H, OCH3) write 3.81 (s, 3H, OCH3).

Line 282, 295, 307instead Intermediate 3 write Intermediate 3i.

Author Response

Dear sir or madam,

      thank you very much for your valuable comments on my article, which is very helpful for my article improvement. I have made any changes or responses to your suggestion, as follows:

Point 1: In this paper the authors describe the procedure for preparation of new 5-aryl-4,5-dyhidrotetrazolo[1,5-a]thieno[2,3-e]pyridines. All target compounds were synthesized for the first time and were well characterized using spectroscopic methods. Pharmacological tests and computation studies were performed to elucidate the antidepressant activity of these compounds. I would suggest a native English speaker to check the manuscript throughout as multiple language errors are still present.

Response 1: we have already done a language retouching of the article through a special English retouching agency.

Point 2: I recommend that present manuscript can be accepted after mayor revision. The following comments are addressed:   In the part 2.1 Chemistry it would be appropriate to extended details and provide additional references for the synthesis of intermediates 2a-p and 3a-p as well as for the target compounds 4a-p, 5, 6 and 7.  

Response 2: In the part 2.1 Chemistry, we have appropriately extended the synthetic details of the intermediates 2a-p, 3a-p and the target compounds 4a-p, 5, 6 and 7, and provided references on similar synthetic methods for the intermediates 2a-p and 3a-p.    

Point 3: Line 71: It would be appropriate to add discussion and reference about Fluoxetine as a positive control.

Response 3: We have been appropriate added some discussion and reference about Fluoxetine in the paper.

Point 4: Line 80: The authors concluded from the Table 1 that compound 4i showed the best activity from the tested compounds. However, compound 4f showed similar activity. It would be convenient to do additional pharmacological tests with this compound (TST, Open-field test and 5-HT concentration determination) as well as docking study.

Response 4: Thank you very much for your suggestion. The FST model is an important animal model for antidepressants screening. According to the FST test, we selected the best antidepressant compound 4i for various pharmacological tests to further explore the possible mode of action of depression activity. In this paper, although compound 4f also has a good antidepressant activity in the FST model, it is slightly weaker than 4i, and because 4f and 4i have little change in structure, only electron-withdrawing F and CF3 are different in structure. They belong to a similar class of compounds, so the way they exert antidepressant effects may be similar. Therefore, we have selected only one representative and most active compound 4i for TST, Open-field test, 5-HT concentration determination, and docking study.

Point 5: In the section 3.1 Chemistry the authors should be provide explanation about starting materials (commercial availability or references for their synthesis). After inspection using SciFinder I found that starting materials 1a-p as well as intermediates 2a-p and 3a-p are not known compounds in literature. For these compounds the authors should provide spectral characterizations such as for the target compounds 4a-p, 5, 6 and 7. These data should be add in the manuscript or in the Supporting material.

Response 5: Thank you very much for your suggestion, the preparation of the starting material 1a-p was carried out by a similar synthesis method of the reference (Zhang, et al. Chin. J. Org. Chem., 2008, 28, 881-884.), and we have added the reference in the paper. The 1H and 13C spectra and analytical data of compounds 1a-p have been attached to the supplymentary. For the synthesis of the intermediates 2a-p and 3a-p, we have not purified, but after the simple processing, the obtained crude product is directly put into the next step, while the intermediates 2a-p and 3a-p have poor stability for a long time and are easily deteriorated. Therefore, no analytical data can be provided; However, the synthetic method reference similar to the intermediate 2a-p, 3a-p has been added in the 2.1. Chemistry section of Results and Discussion.

Point 6: The following comments should be also addressed:

It is more convenient to use the name 5-aryl-4,5-dyhidrotetrazolo[1,5-a]thieno[2,3-e]pyridine instead 5-substituted phenyl-4,5-dyhidrotetrazolo[1,5-a]thieno[2,3-e]pyridine. These corrections should be applied in lines: 3, 12, 37, 179 and 346.  

 Line 15 ─ Delete: The results of...

Line 19 ─ Replace 5-HT with 5-hydroxytriptamine (5-HT).

Line 31 ─ Replace that tetrazole with that compounds with tetrazole.

Line 44 ─ Instead 4-phenylpiperidine write 4-arylpiperidine.

Line 56, 57 ─ Explanation in brackets is not clear!!!

Line 67 ─ It would be more convinient to write subtitle: 2.2.1. Forced swiming test (FST) Line 84─ Delete: To our disappointed...

Line 101 ─ It would be more convinient to write subtitle: 2.2.2. Tail suspension test (TST)

Line 164─ Instead IRPrestige write IR Prestige (Typographical errors).

Line 168─ It something wrong with this procedure. It is impossible to add 100 mL of a solvent into a 100 mL round bottom flask. Some correction is necessary.

172─ Please add reaction time.

Line 172─ Instead extracted write washed.

Line 178─ Instead methanol cooled write cold methanol.

Line 181─ It more appropriate instead and reduce the reaction system to 0-5oC write and cool the mixture in an ice water.

Response 6: All of above work has been done. and we have already explained "Line 56, 57 ─ Explanation in brackets is not clear!!!" in the paper.

Point 7: General instruction for interpretation of NMR spectra:

Give 1H NMR chemical shifts to two digits after the decimal point. Include number of protons represented by the signal, peak multiplicity, and coupling constants needed. (J italicized, reported with up to one digit after the decimal).

Give 13C NMR chemical shifts to one digit after the decimal point, unless an additional digit will help distinguish overlapping peaks.

Response 7: Thank you very much for your suggestion, this work has been done.

Point 8:  Line 199, 235, 253, 259─wrong interpretation of 1H NMR spectra. Instead a duble of dublet 1,4 disubstituted aromatic compounds have two separated doublets. For example, instead 7.16-7.35 (dd, 4H) it is 7.19 (d, 2H) and 7.35 (d, 2H) with appropriate coupling constants.

Line 240─ wrong interpretation of 1H NMR spectra. Instead 3.66-3.68 (d, 2H, J = 7.23 Hz, CH2) write 3.67(d, 2H, J = 7.23 Hz, CH2). Also, instead 3.82 (s, 1H, OCH3) write 3.82 (s, 3H, OCH3). Line 246─ instead 3.80 (s, 1H, OCH3) write 3.80 (s, 3H, OCH3).

Line 252─ instead 3.81 (s, 1H, OCH3) write 3.81 (s, 3H, OCH3).

Line 282, 295, 307─instead Intermediate 3 write Intermediate 3i.

Response 8: Thank you very much for your suggestion, this work has been done.

Reviewer 3 Report

Manuscript present sound and comprehensive data on synthesis and antidepressant activity of thienopyridine analogs of paroxetine. Authors described chemistry, biological activity in mouse model and docking study in silico.  However some more general conclusion about plausible mechanism of action is missing. Particularly the docking studies are described very briefly.

Then the manuscript needs some language and style corrections few examples below:

“ To our disappointed, the” - To our disappointment

“obtained the active structure-activity relationship” – remove “active”

“Therefore, the antidepressant activity of compound 4i has dose dependent.” – what? Is seems there is missing something “ratio? Shape?”

To sum up I would recommend acceptance after minor revision

Author Response

Dear sir or madam,

    thank you very much for your valuable comments on my article, which is very helpful for my article improvement. I have made any changes or responses to your suggestion, as follows:

Point 1: Manuscript present sound and comprehensive data on synthesis and antidepressant activity of thienopyridine analogs of paroxetine. Authors described chemistry, biological activity in mouse model and docking study in silico.  However some more general conclusion about plausible mechanism of action is missing. Particularly the docking studies are described very briefly.

Response 1: Thank you very much for your suggestion, we have been appropriate added some general conclusion about plausible mechanism of action for docking studies in the “2.2.5. Docking study” part.

Point 2: Then the manuscript needs some language and style corrections few examples below:

“ To our disappointed, the” - To our disappointment.

“obtained the active structure-activity relationship” – remove “active”

Response 2: This work has been done, and we have already done a language retouching of the article through a special English retouching agency.

Point 3: “Therefore, the antidepressant activity of compound 4i has dose dependent.” – what? Is seems there is missing something “ratio? Shape?” To sum up I would recommend acceptance after minor revision

Response 3: Thank you very much for your suggestion. I have described it incorrectly. This sentence has been revised in the text as: Therefore, the antidepressant activity of compound 4i appeared to be dose-dependent, with a gradually increased antidepressant activity observed when increasing the dose.

Round  2

Reviewer 1 Report

I have no other comments.

Reviewer 2 Report

The authors have addressed all the issues raised during initial assessment, so I recommend the revised manuscript for publishing.